# Mapping and Candidate Gene Analysis of an All-Stage Stem Rust Resistance Gene in Durum Wheat Landrace PI 94701

**DOI:** 10.3390/plants13162197

**Published:** 2024-08-08

**Authors:** Hongyu Li, Kairong Li, Hongna Li, Chen Yang, Geetha Perera, Guiping Wang, Shikai Lyu, Lei Hua, Shams ur Rehman, Yazhou Zhang, Michael Ayliffe, Haitao Yu, Shisheng Chen

**Affiliations:** 1National Key Laboratory of Wheat Improvement, School of Advanced Agricultural Sciences, Peking University, Beijing 100871, China; hongyuli8@163.com; 2National Key Laboratory of Wheat Improvement, Peking University Institute of Advanced Agricultural Sciences, Shandong Laboratory of Advanced Agriculture Sciences in Weifang, Weifang 261325, China; kairongli@163.com (K.L.); hongna.li@pku-iaas.edu.cn (H.L.);; 3College of Agronomy, Shandong Agricultural University, Taian 271018, China; 4Triticeae Research Institute, Sichuan Agricultural University, Chengdu 611130, China; 5CSIRO Agriculture and Food, GPO Box 1700, Clunies Ross Street, Canberra, ACT 2601, Australia; 6Wheat Research Institute, Weifang Academy of Agricultural Sciences, Weifang 261071, China

**Keywords:** durum wheat, stem rust, resistance gene, *SrPI94701*, BSR-Seq, CC-NBS-LRR

## Abstract

*Puccinia graminis* f. sp. *tritici* (*Pgt*), the causal agent of wheat stem rust, poses a significant threat to global wheat production. Genetic resistance offers a cost-effective and sustainable solution. The durum wheat landrace PI 94701 was previously hypothesized to carry two stem rust resistance (*Sr*) genes, but their chromosomal locations were unknown. In this study, we mapped and characterized an all-stage *Sr* gene in PI 94701, temporarily designated as *SrPI94701*. In seedling tests, *SrPI94701* was effective against all six *Pgt* races tested. Using a large segregating population, we mapped *SrPI94701* on chromosome arm 5BL within a 0.17-cM region flanked by markers *pku69124* and *pku69228*, corresponding to 1.04 and 2.15 Mb genomic regions in the Svevo and Chinese Spring reference genomes. Within the candidate region, eight genes exhibited differential expression between the *Pgt*-inoculated resistant and susceptible plants. Among them, two nucleotide-binding leucine-rich repeat (NLR) genes, *TraesCS5B03G1334700* and *TraesCS5B03G1335100*, showed high polymorphism between the parental lines and were upregulated in *Pgt*-inoculated resistant plants. However, the flanking and completely linked markers developed in this study could not accurately predict the presence of *SrPI94701* in a survey of 104 wheat accessions. *SrPI94701* is a promising resource for enhancing stem rust resistance in wheat breeding programs.

## 1. Introduction

Wheat stem rust, a damaging fungal disease caused by *Puccinia graminis* f. sp. *tritici* (*Pgt*), has historically caused significant yield losses in wheat worldwide. In China, wheat stem rust currently occurs sporadically under natural conditions [1]. Historically, it has been a major problem in wheat production across Africa, Asia, Australia, Europe, and the Americas [2]. Losses are often severe, ranging from 50% to 90%, particularly when there is substantial disease pressure during grain filling. In some instances, individual fields can be destroyed in regions conducive to disease development, making it impossible to grow susceptible cultivars [3,4]. Growing resistant wheat cultivars stands as the most cost-effective and environmentally sustainable method for controlling this disease [5].

Highly virulent races of *Pgt* have emerged, overcoming numerous widely deployed resistance genes present in elite wheat cultivars globally. In 1998, a *Pgt* race, TTKSK (also known as Ug99), was identified in Uganda, exhibiting virulence against the widely deployed *Sr* genes, *Sr31* and *Sr38* [6]. Subsequent to the deployment of *Sr24* in Kenya [7], further selection for virulence resulted in the identification of races TTKST, TTTSK (*Sr36* virulence) [8], and more recently, TTKTT and TTKTK (*SrTmp* virulence) [9]. To date, a total of 15 variants within the Ug99 race group have been documented across at least 13 countries, highlighting the adaptability and widespread dissemination of *Pgt* [10,11]. Recently, Ug99 has been detected in Nepal [12], posing a threat to wheat production in South and Eastern Asia. In addition to the Ug99 lineage, several other highly virulent *Pgt* races have also been identified in stem rust outbreaks, including JRCQC, TKTTF, and TRTTF. The non-Ug99 race JRCQC was identified in Ethiopia [13]. This race exhibits combined virulence against *Sr9e* and *Sr13b*, which are commonly deployed in durum wheat [13,14]. Race TKTTF caused severe stem rust epidemics in Ethiopia during 2013–2014, affecting the widely grown Ug99-resistant wheat cultivar “Digalu” [15]. Subsequently, TKTTF was detected in at least ten countries [16,17]. Race TRTTF was observed in various countries, including Iran, Yemen, Ethiopia, and Pakistan [18,19]. This race defeated the resistance conferred by *Sr* genes *Sr36*, *SrTmp*, and *Sr1RS*^Amigo^, which are effective against the original Ug99 race TTKSK [13]. Additionally, novel *Pgt* pathotypes caused severe wheat stem rust epidemics and outbreaks in Central Asia [20]. This continuous evolution of the pathogen highlights the ongoing necessity to explore new effective *Pgt* resistance genes for wheat breeding programs.

To date, more than 60 stem rust resistance genes have been formally cataloged in bread and durum wheats, as well as their wild relatives [21,22]. Among these, tetraploid wheat (*Triticum turgidum* ssp.) has contributed multiple *Sr* genes (or alleles), such as *Sr2*, *Sr9d*/*Sr9e*/*Sr9g*, *Sr11*, *Sr12*, *Sr13a*/*Sr13b*/*Sr13c*/*Sr13d*, *Sr14*, and *Sr17* [4,14,23,24,25]. All of these genes from tetraploid wheat are all-stage resistance (ASR) genes, with the exception of *Sr2*, which is an adult plant resistance (APR) gene [26]. ASR genes typically confer a high level of resistance at all stages of plant growth and development, but they are often specific to certain pathogen races. In contrast, APR genes tend to provide only partial resistance at later stages of plant development, and a single APR gene is often inadequate for achieving satisfactory levels of plant protection [27]. Due to the large and complex nature of the polyploid wheat genome, only two ASR *Sr* genes (*Sr9* and *Sr13*) have been successfully cloned so far in durum wheat [14,25]. As durum and bread wheat share common A and B genomes, it is relatively easy to transfer important *Sr* genes from *T. durum* into bread wheat, allowing these genes to be utilized in bread wheat breeding programs.

The durum wheat landrace PI 94701, collected in Palestine, was previously postulated to carry two temporarily designated *Sr* genes (*Srdp1* and *Srdp2*) based on the observed segregation of resistance against *Pgt* race 11-SS2 [28]. Plants carrying only *Srdp1* displayed resistant infection types (ITs = 0; to 1+), and plants carrying only *Srdp2* showed moderate resistance (ITs = 1+ to 2−) [28]. Initially, *Srdp2* was proposed to be allelic to *Sr13*, a gene located on chromosome arm 6AL. However, subsequent studies revealed that *Srdp2* is distinct from *Sr13* based on pathogenic variability investigations and sequencing of polymerase chain reaction (PCR) products of diagnostic markers for *Sr13* [14]. Despite these findings, the chromosomal locations of *Srdp1* and *Srdp2* remain unknown. The primary objectives of the present study were to: (1) assess whether PI 94701 confers resistance to Chinese *Pgt* races; (2) genetically map the stem rust resistance genes present in PI 94701; and (3) identify the corresponding regions within sequenced wheat genomes and predict potential candidate genes.

## 2. Results

### 2.1. Characterization of Stem Rust Resistance in Durum Wheat Accession PI 94701

Durum wheat accessions PI 94701 and Rusty were challenged with six *Pgt* races, comprising five from China and one from North America (Appendix A). PI 94701 displayed high resistance [infection types (ITs) = 1 or 1+] to all tested *Pgt* races, whereas Rusty exhibited susceptible infection types ranging from 3+ to 4 (Figure 1). The F_1_ plants resulting from the cross between PI 94701 and Rusty exhibited resistance against *Pgt* race 34MKGQM. A subset of 143 F_2_ plants derived from this cross was evaluated for their reactions to *Pgt* race 34MKGQM. Within this subset, 105 plants showed resistance (ITs = 1 to 2−), while 38 plants were susceptible (ITs = 3 to 4). Chi-squared analysis of the phenotyping data revealed a segregation ratio of 3:1 consistent with a single dominant gene (*χ*^2^ = 0.19, *p* = 0.66), which is henceforth referred to as *SrPI94701*. Among the 143 F_2:3_ families evaluated with *Pgt* race 34MKGQM, 32 were homozygous resistant, 73 were segregating, and 38 were homozygous susceptible. This distribution corresponded to the expected 1:2:1 segregation ratio (*χ*^2^ = 0.57, *p* = 0.75).

Based on their resistance or susceptibility to *Pgt* race 34MKGQM, ten homozygous resistant and ten homozygous susceptible F_2:3_ families were selected from the PI 94701 × Rusty cross. These families were further tested against five additional *Pgt* races: BCCBC, 34MTGSM, 21C3CTTTM, 34C3RTGQM, and 34C3RKGQM, and were grown in five separate growth chambers. The ten families exhibiting resistance to 34MKGQM also demonstrated robust resistance to other *Pgt* races, while the ten 34MKGQM-susceptible families showed consistent susceptibility (Appendix A). These results suggest that the resistance of PI 94701 to races BCCBC, 34MTGSM, 21C3CTTTM, 34C3RTGQM, and 34C3RKGQM is linked to the *SrPI94701* genomic region.

### 2.2. Genetic Mapping of SrPI94701 on Chromosome Arm 5BL

For the initial mapping, bulked segregant RNA-Seq (BSR-seq) was performed on the F_2:3_ mapping population evaluated with *Pgt* race 34MKGQM, revealing 32,708 single nucleotide polymorphisms (SNPs) between PI 94701 and Rusty. Based on SNP index analysis, 160 SNPs within the genomic region spanning from 673.5 to 699.9 Mb (Svevo Rel.1.0) on the long arm of chromosome 5B exhibited significant association with the phenotype (Figure 2, Appendix A). This result indicates that *SrPI94701* was located on chromosome arm 5BL.

To verify the mapping location, we specifically selected SNPs surrounding the region of interest on chromosome arm 5BL and developed 14 5B-genome-specific Cleaved Amplified Polymorphic Sequence (CAPS) markers (Table 1). Subsequently, these markers were used to genotype the 143 F_2:3_ families, resulting in the construction of a genetic linkage map for *SrPI94701* (Figure 3A,B). Based on the linkage results, *SrPI94701* was found to be completely linked to marker *pku69187* and mapped within a 1.4 cM interval flanked by markers *pku69119* and *pku69231* (Figure 3B).

To precisely determine the position of *SrPI94701*, an additional 1008 F_4_ plants from six selected segregating F_3_ families were screened for recombinants between markers *pku69119* and *pku69231*. This screening identified ten plants carrying recombination events between these two markers. Using these ten recombinants detected in this screening and one previously identified recombinant in 143 F_3_ families, the genetic distance between markers *pku69119* and *pku69231* was recalculated to be 0.48 cM (Figure 3C). We evaluated the progeny of these plants carrying informative recombination events with *Pgt* race 34MKGQM. Using these recombinants and four new PCR markers developed within this genomic region (Table 1), *SrPI94701* was further mapped to a 0.17-cM interval flanked by CAPS markers *pku69124* and *pku69228*, and was completely linked to markers *pku69187*, *pku69211*, and *pku69227* (Figure 3C). PCR amplifications with six CAPS markers are showcased in Figure 4.

### 2.3. Candidate Genes for SrPI94701 within Tetraploid and Hexaploid Wheat Genomes

The 0.17-cM candidate region delimited by PCR markers *pku69124* and *pku69228* spans a 1.04 Mb genomic region in the *T. durum* reference genome of Svevo (691.24–692.28 Mb, Figure 3D) and a 2.15 Mb genomic region in the *T. aestivum* reference genome of Chinese Spring (CS; 702.56–704.71 Mb). These candidate regions contain 65 annotated genes in Svevo (*TRITD5Bv1G248710*–*TRITD5Bv1G249360*; Appendix A) and 28 high-confidence genes in CS (*TraesCS5B03G1334300*–*TraesCS5B03G1340200*; Appendix A). These annotated genes include one typical NLR gene in Svevo and seven in CS, which is particularly noteworthy for this study as NLRs are the predominant class of genes associated with disease resistance in plants [25,32,33,34]. In addition to these NLR genes, we identified one protein kinase in Svevo and six in CS (Appendix A) as potential candidates due to their known roles in disease resistance across various plant species [35,36,37,38,39].

Subsequently, we analyzed the transcript levels of the candidate genes using published RNAseq studies compiled in the wheat ExpVIP database (http://www.wheat-expression.com/; accessed on 8 June 2024). Of the 28 high-confidence genes annotated in the candidate gene region in the CS genome, only 16 high-confidence genes were found to be expressed in wheat leaves (Appendix A).

### 2.4. Identification of Differentially Expressed Genes (DEGs) Within the SrPI94701 Mapping Interval

Transcript levels of the candidate genes were analyzed using RNA-seq data from the F_4_ sister lines S73 and R32, selected from the PI 94701 × Rusty cross. Given the identification of numerous NLR and kinase genes in the colinear regions of CS (Appendix A), we focused on analyzing the 28 annotated genes within the candidate interval in CS. Among these candidate genes, eight exhibited differential expression between the *Pgt*-inoculated resistant and susceptible plants (Figure 5). Seven DEGs showed significant upregulation in R32 plants compared to S73 plants, whereas only one DEG displayed downregulation (Appendix A). These upregulated DEGs included four NLR genes: *TraesCS5B03G1334700*, *TraesCS5B03G1335000*, *TraesCS5B03G1335100*, and *TraesCS5B03G1335600*, making them strong candidate genes.

We used whole-genome resequencing reads from PI 94701 and Rusty to assemble genomic contigs containing the identified NLR DEGs. Upon comparison of these contigs between PI 94701 and Rusty, we observed that these NLRs were either missing or partially deleted in Rusty (Appendix A). Among them, we excluded *TraesCS5B03G1335000* and *TraesCS5B03G1335600* as candidate genes due to the identical nature (100% match) of their coding and promoter regions between PI 94701 and CS. However, CS exhibited susceptibility to *Pgt* races 34MTGSM, 34C3RTGQM, 21C3CTTTM, 34MKGQM, and 34C3RKGQM at the seedling stage, suggesting the absence of *SrPI94701*.

To validate the RNA-seq findings, we measured the expression levels of the candidate genes *TraesCS5B03G1334700* and *TraesCS5B03G1335100* relative to *ACTIN* using quantitative real-time PCR (qRT-PCR). The results confirmed a significant upregulation (*p* < 0.05) of these candidate genes between the two groups (Appendix A), supporting the conclusions from the RNA-seq analysis.

### 2.5. Validation of SrPI94701-Linked Markers in Uncharacterized Wheat Accessions

To assess the significance of the haplotype defined by the two flanking and three completely linked markers (Figure 3C and Figure 4), we used these markers to evaluate 53 *T. dicoccon* and 51 *T. aestivum* accessions. Among the 53 *T. dicoccon* genotypes, 29 accessions (54.7%) exhibited an identical haplotype to the resistant parent, PI 94701 (Appendix A). Nevertheless, certain accessions with the *SrPI94701* haplotype, like PI 273980 and PI 197489, were susceptible to *Pgt* race 34MKGQM (Appendix A), indicating that these markers could not accurately predict the presence of *SrPI94701* in uncharacterized *T. dicoccon* genotypes.

Among the 51 tested hexaploid wheat accessions, the *SrPI94701* haplotype was detected in 8 accessions (15.7%; Appendix A). Similarly, two accessions carrying the *SrPI94701* haplotype (Huaimai40 and Luyuan502) showed susceptibility to *Pgt* race 34MKGQM, suggesting that these markers were ineffective in predicting the presence of *SrPI94701* in hexaploid wheat.

## 3. Discussion

Initially, the durum wheat landrace PI 94701 was reported to carry a single *Sr* gene conferring resistance to *Pgt* race 15B [40]. Later, it was proposed that PI 94701 carries two provisionally designated *Sr* genes (*Srdp1* and *Srdp2*) based on the segregation of resistance against *Pgt* race 11-SS2 [28]. However, the precise chromosomal locations of *Srdp1* and *Srdp2* remained unknown. In this study, using Chinese *Pgt* races, we successfully mapped the stem rust resistance gene *SrPI94701* within a 0.17 cM region on the long arm of chromosome 5B. However, the lack of chromosomal locations and genetic maps for *Srdp1* and *Srdp2* has limited our ability to establish the mapping relationship between *SrPI94701* and *Srdp1*/*Srdp2*.

The clustering of NLR genes promotes the generation of new variants through duplication, recombination, and conversion events, thereby increasing the diversity of NLR genes [14,41]. Using the reference genomes of tetraploid and hexaploid wheat, we defined the *SrPI94701* candidate region as a 1.04 Mb region in *T. durum* wheat Svevo and a 2.15 Mb region in hexaploid wheat CS, which includes a cluster of typical NLR genes. However, NLR genes identified within the colinear regions of the Svevo and CS reference genomes display copy number variations (Appendix A). Similar to the *SrPI94701* candidate region, reports of copy number and structural variations of NLR genes have been documented for previously isolated disease resistance genes in wheat and its wild relatives [14,25,33,41,42]. Considering that many cloned *R* genes in plants encode intracellular NLR proteins [32,34], these NLR genes within the mapped region are strong candidates for *SrPI94701*.

In addition to the NLR genes, several protein kinases were also identified in the *SrPI94701* mapping region (Appendix A). This gene family is frequently linked to disease resistance in various plant species [43,44]. In wheat, six cloned rust resistance genes, *Sr60* [35], *Sr62* [45], *Sr43* [46], *Yr36* [36], *Yr15* [39], and *Lr9*/*Lr58* [47], encode proteins with kinase domains. These NLRs and protein kinases have been prioritized for functional characterization to explore their potential as the causal gene for *SrPI94701*. However, we cannot rule out the possibility of additional NLRs or protein kinases present in PI 94701 that are absent in the reference genomes.

The wheat stem rust resistance genes, *Sr49* and *Sr56*, were previously mapped close to *SrPI94701* on the long arm of chromosome 5B [48,49]. *Sr56*, known for conferring adult plant resistance, was mapped in the Swiss winter wheat cultivar Arina and demonstrated a 12–15% reduction in stem rust severity under field conditions [49]. *Sr56* was flanked by SSR markers *sun209* and *sun215* [49], corresponding to a genomic region ranging from 705,196,548 bp to 705,764,022 bp in the CS RefSeq v2.1 reference genome. These results suggest that *SrPI94701* is distinct from the *Sr56* gene. The ASR gene *Sr49* was mapped in the hexaploid wheat landrace AUS28011 and showed effectiveness against all commercially important Australian *Pgt* races that were examined, displaying mesothetic resistant (ITs = 2−) infection types [48]. The location of *Sr49* was delimited by SSR markers *sun479* and *sun209*, spanning a genetic region from 702,174,220 bp to 705,196,462 bp (CS RefSeq v2.1) which includes our candidate gene region (702.56–704.71 Mb, Appendix A). However, there are three differences between *SrPI94701* and *Sr49*. First, *SrPI94701* was discovered in durum (tetraploid) wheat, while *Sr49* was identified in hexaploid wheat. Second, wheat plants carrying *SrPI94701* (ITs = 1 or 1+) showed better resistance compared to those with *Sr49* (ITs = 2−). Third, AUS28011 (*Sr49*) and PI 94701 (*SrPI94701*) showed different haplotypes when genotyped with the flanking and completely linked markers of *SrPI94701*. Nonetheless, due to their co-location within overlapping genetic intervals, *SrPI94701* and *Sr49* may be the same gene.

Previously, no haplotype containing both flanking marker alleles *sun479* and *sun209* was identified in the screening of 152 Australian and Nordic wheat lines. Nevertheless, six wheat lines carried the *Sr49*-linked *sun479_200bp_* allele, and seven harbored the *Sr49*-linked *sun209_148bp_* allele [48]. The *Sr49* locus was also detected in a Genome-Wide Association Study (GWAS) involving a panel of 283 durum wheat lines challenged with various East African stem rust isolates [50]. These two markers (*sun479* and *sun209*) were detected with a minor allele frequency (average of 0.053), indicating the rarity of *Sr49* within this germplasm panel [50]. In this study, the genotypes of the flanking and completely linked markers of *SrPI94701* revealed that 54.7% of *T. dicoccon* and 15.7% of *T. aestivum* accessions showed the *SrPI94701*-linked resistant haplotype. The notable differences in allele frequencies observed could be attributed to the diversity among the germplasm sources. Alternatively, it is also possible that they represent distinct genes, which explains the obvious differences in allele frequencies detected.

*SrPI94701* confers resistance against multiple *Pgt* races in China, making it a potentially valuable genetic resource in wheat breeding. However, as *SrPI94701* provides only an intermediate level of resistance when used alone, it needs to be combined with other *Sr* genes to provide economically useful levels of resistance. Some potentially valuable *Sr* genes that could enhance resistance include *Sr35* [51], *Sr13* [14], *Sr22a*/*Sr22b* [52,53], and *Sr26* [54]. Despite this limitation, *SrPI94701* can provide an additional layer of resistance against various *Pgt* races and contribute to diversifying the sources of resistance used in wheat breeding programs.

## 4. Materials and Methods

### 4.1. Plant Materials and Mapping Populations

The *Pgt*-resistant *Triticum turgidum* ssp. *durum* (2*n* = 4*x* = 28, AABB) wheat accession PI 94701 was crossed with the highly susceptible durum wheat genotype Rusty [14,55]. For the initial mapping, we evaluated a population of 143 F_2_ plants derived from the PI 94701 × Rusty cross inoculated with Chinese *Pgt* race 34MKGQM (isolate 20IAL06). To construct a high-density genetic map, six F_2_ plants (plants 15R, 37R, 75R, 90R, 118R, and 126R) heterozygous for the *SrPI94701* region, identified using flanking molecular markers, were selected. These plants generated a secondary population consisting of 1008 F_3_ individuals. These F_3_ plants were genotyped with *SrPI94701* flanking markers to identify plants with recombination events within the candidate gene region. The recombinants and their corresponding F_4_ progeny (~25 plants per family) were challenged with *Pgt* race 34MKGQM. As a source of *Sr49*, we used the bread wheat landrace AUS28011 (Mahmoudi) from Tunisia, which exhibited good resistance to five Australian *Pgt* pathotypes [48]. Finally, we used a collection of 53 accessions of *T. turgidum* ssp. *dicoccon* and 51 accessions of *T. aestivum* to determine the value of the *SrPI94701*-linked markers developed in the present study for marker-assisted selection.

### 4.2. Evaluation for Stem Rust Resistance

Stem rust seedling assays for both the parental lines and the segregating populations were conducted at the Peking University Institute of Advanced Agricultural Sciences following previously described procedures [56,57]. Six Chinese and North American *Pgt* races, namely BCCBC, 34MTGSM, 34C3RTGQM, 21C3CTTTM, 34MKGQM, and 34C3RKGQM [58], were evaluated for virulence against parental lines PI 94701 and Rusty. The avirulence/virulence profiles of these *Pgt* races are detailed in Appendix A. *Pgt* urediniospores, stored at − 80 °C, were subjected to heat shock at 42 °C for 3 min. Subsequently, a urediniospore/talc suspension was applied to seedlings at the three-leaf stage using the shaking-off method [29]. Following inoculation, the plants were kept in a dark dew chamber overnight and then transferred back to growth chamber conditions maintained at 22–24 °C with a 16 h photoperiod. *Pgt* infection types (ITs) were recorded at 12–14 dpi using the Stakman 0–4 scale, indicating levels of immunity to susceptibility [59]. For plants carrying recombination events within the candidate region, progeny tests were conducted using approximately 25 plants from each F_2:3_ family.

### 4.3. Bulked Segregant RNA-Seq (BSR-Seq) Analysis

Based on the phenotypic data against *Pgt* race 34MKGQM, 15 homozygous resistant and 15 homozygous susceptible F_3_ families were selected to constitute the resistant bulk (R-bulk) and susceptible bulk (S-bulk) for RNA extraction. Leaves of equal length were taken from the second leaf of all seedlings in each family. After harvest, each sample was immediately flash-frozen in liquid nitrogen and stored at −80 °C for RNA extraction. Total RNAs were extracted from the resistant parent PI 94701 and R/S-bulks using the Spectrum^TM^ Plant Total RNA Kit (MilliporeSigma, Burlington, MA, USA). RNA sequencing was conducted at Novogene Bioinformatics Technology Co., Ltd. (Beijing, China) with 150-base paired-end reads. The raw sequencing data are available at the National Genomics Data Center (NGDC) under the BioProject accession number PRJCA027291. For the susceptible parent Rusty, we used the whole-genome resequencing data described in Wang et al. (2023a) under the BioProject accession number PRJCA017761.

Raw RNA-seq reads were trimmed using Trimmomatic v. 0.32 [60] to remove low-quality nucleotide sequences. The trimmed reads were then aligned to the reference genome of the durum wheat cultivar Svevo [61] using STAR v2.5.0c [62]. SNPs were identified using the HaplotypeCaller tool from GATK v3.2-2 [63]. SNP-index and Δ(SNP-index) were calculated across the whole genome to identify potential regions of interest using the methods described previously [30,64,65].

### 4.4. Development of PCR Markers

To accelerate the development of PCR markers, we performed whole-genome resequencing for the resistant parent PI 94701 (accession number PRJCA027291). PCR primers were designed using the Primer3 software (https://bioinfo.ut.ee/primer3-0.4.0/primer3/; accessed on 8 June 2024). The identified SNPs within the mapping interval were used to develop CAPS markers [66]. Enzyme cleavage reactions of the CAPS markers were performed according to standard procedures for the corresponding restriction endonucleases (New England Biolabs, Hitchin, UK). PCR reactions were conducted using a Veriti 96-Well Fast Thermal Cycler (Thermo Fisher Scientific Inc., Waltham, MA, USA). After the PCR amplification, 10 µL of PCR products were digested with the suitable restriction enzyme. Subsequently, 5 µL of digested PCR products were analyzed using agarose gel electrophoresis, and the gels were stained with ethidium bromide.

### 4.5. qRT-PCR Analysis

From the PI 94701 × Rusty population, a pair of F_4_ sister lines homozygous for the absence (S73) or presence (R32) of *SrPI94701* were developed using PCR markers. The susceptibility/resistance responses for S73 and R32 against *Pgt* race 34MKGQM were evaluated. Leaves from S73 and R32 inoculated with race 34MKGQM were sampled at 6 dpi, with three biological replicates used for each genotype. RNA sequencing was conducted at Novogene Bioinformatics Technology Co., Ltd. (Beijing, China). The raw sequencing data are available under the same BioProject accession number, PRJCA027291. DEGs between the sister lines S73 and R32 were determined using the edgeR software v. 4.2.1, applying significance thresholds of FDR < 0.05, *p*-value < 0.05, and |log2foldchange| > 1 [67]. DEGs within the *SrPI94701* candidate region were identified, and a heatmap was generated using the pheatmap R package (https://cran.ms.unimelb.edu.au/web/packages/pheatmap/pheatmap.pdf; accessed on 8 June 2024). qRT-PCR reactions were performed using an ABI QuantStudio 5 Real-Time PCR System (Applied Biosystems, Foster City, CA, USA). Transcript levels were determined in four biological replicates and quantified as fold-*ACTIN* levels using the 2^−ΔCT^ method [33,41].

### 4.6. Statistical Analyses

Genetic linkage maps with polymorphic markers and stem rust resistance scores of F_2:3_ families were constructed using the MapChart v2.2 software (https://www.wur.nl/en/show/Mapchart.htm; accessed on 8 June 2024) [31]. The significance of the differences in transcript levels was evaluated using a two-sided, unpaired *t*-test.

## 5. Conclusions

In this study, we mapped and characterized *SrPI94701*, an all-stage stem rust resistance gene in durum wheat landrace PI 94701. The broad efficacy of *SrPI94701* in China makes it a desirable target for introgression into Chinese bread wheat cultivars. For recurrent wheat cultivars with different marker haplotypes, the flanking and fully linked markers for *SrPI94701* can be used to transfer this gene from durum into modern bread wheat varieties. While *SrPI94701* confers only partial resistance to *Pgt* races, this might be beneficial in breeding programs aiming to combine multiple partial resistance genes and avoid major all-stage resistance genes, a strategy that has been proposed to increase the durability of wheat resistance to stem rust [68].

Using a large mapping population consisting of 2302 recombinant gametes, *SrPI94701* was successfully mapped to the telomeric region of chromosome arm 5BL within a 0.17-cM region. The presence of NLR genes in the collinear regions of wheat reference genomes indicates that *SrPI94701* may be a typical NLR gene. However, final identification of the *SrPI94701* gene will be required to fully test this hypothesis and determine the relationship between *SrPI94701* and *Sr49*. To ascertain if the two identified NLR candidate genes are required for resistance to *Pgt*, we are currently conducting functional characterization using truncation mutations for each gene.

In summary, the high-resolution genetic map of *SrPI94701* and the closely linked PCR markers developed in this study will facilitate map-based cloning of this *Sr* gene and accelerate its incorporation into modern wheat breeding programs.

## Figures and Tables

**Figure 1 plants-13-02197-f001:**
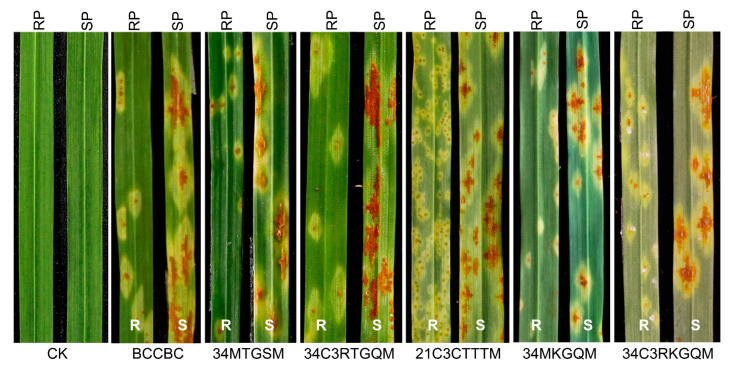
Infection types of PI 94701 and Rusty in response to *Pgt* races. Six *Pgt* races (BCCBC, 34MTGSM, 34C3RTGQM, 21C3CTTTM, 34MKGQM, and 34C3RKGQM) were used in this study, and their avirulence/virulence profiles are detailed in Appendix A. Plants were grown in growth chambers at 22–24 °C with 16 h light/8 h dark. Seedlings at the three-leaf stage were challenged with a urediniospore/talc suspension using the shaking-off method [29]. RP, resistant parent; SP, susceptible parent; R, resistant; S, susceptible; CK, plants without inoculation.

**Figure 2 plants-13-02197-f002:**
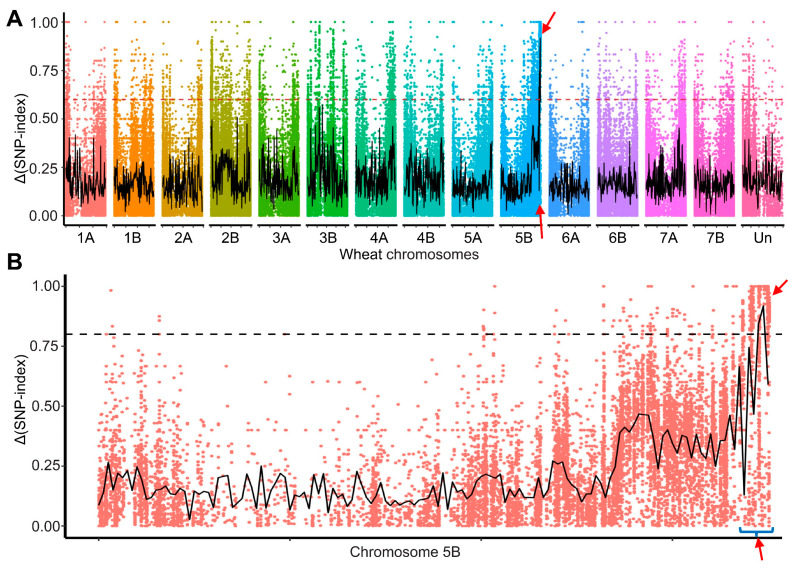
BSR-Seq analysis of *SrPI94701*. (**A**) Δ(SNP-index) was calculated for each SNP across all chromosomes. (**B**) Δ(SNP-index) were specifically calculated for each SNP on chromosome 5B. SNP-index plotting was conducted following a previously established protocol [30]. SNP index values in the resistant and susceptible pools were calculated using a custom Perl script, and the Δ(SNP-index) = |(SNP-index of the resistant pool) − (SNP-index of the susceptible pool)| was determined for each SNP. Sliding window analysis was applied to ΔSNP-index plots with a 5 Mb window size and a 50 kb increment. A total of 160 SNPs within the genomic region spanning from 673.5 to 699.9 Mb (Svevo Rel.1.0; Appendix A) on chromosome arm 5BL showed significant association with the phenotype. The mapping region is highlighted by red arrows.

**Figure 3 plants-13-02197-f003:**
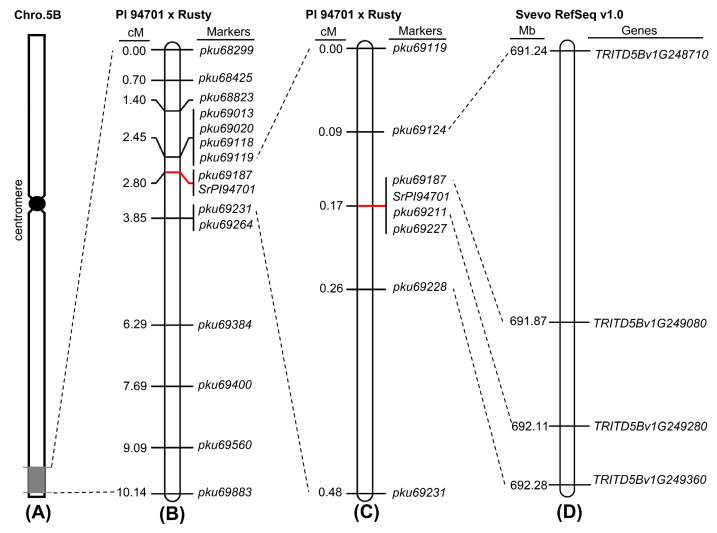
Genetic maps for *SrPI94701*. (**A**) Genetic mapping of the resistance locus on chromosome arm 5BL by BSR-seq. The gray rectangle indicates the genomic region spanning from 673.5 to 699.9 Mb (Svevo Rel.1.0). (**B**) Genetic map for *SrPI94701* based on 143 F_2:3_ families from the PI 94701 × Rusty cross and 14 PCR markers. The values to the left of the markers indicate the genetic distances in centimorgans (cM). (**C**) High-density map based on 1151 F_2_ plants and seven molecular markers. The values to the left of the markers indicate the genetic distances in centimorgans (cM). (**D**) Colinear genomic region on chromosome 5B of Svevo (Rel.v1.0). The values to the left of the genes indicate their physical locations in megabases (Mb). Genetic maps were constructed using the software MapChart v2.2 [31].

**Figure 4 plants-13-02197-f004:**
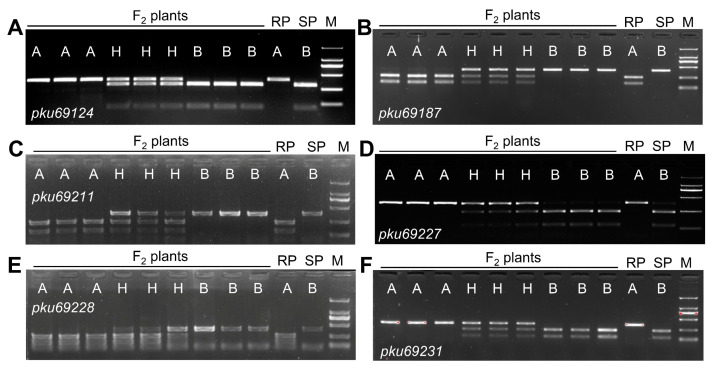
Genotypes of F_2_ plants derived from the PI 94701 × Rusty cross. F_2_ plants genotyped with (**A**) CAPS marker *pku69124* (digested with SspI); (**B**) CAPS marker *pku69187* (BbvCI); (**C**) CAPS marker *pku69211* (HpyCH4IV); (**D**) CAPS marker *pku69227* (PvuII); (**E**) CAPS marker *pku69228* (Hpy188III); and (**F**) CAPS marker *pku69231* (BtgI). PCR products were separated on agarose gels and stained directly with ethidium bromide. RP, resistant parent PI 94701; SP, susceptible parent Rusty; A, band corresponds to the resistant allele; B, band corresponds to the susceptible allele; H, heterozygous; M, DNA ladder.

**Figure 5 plants-13-02197-f005:**
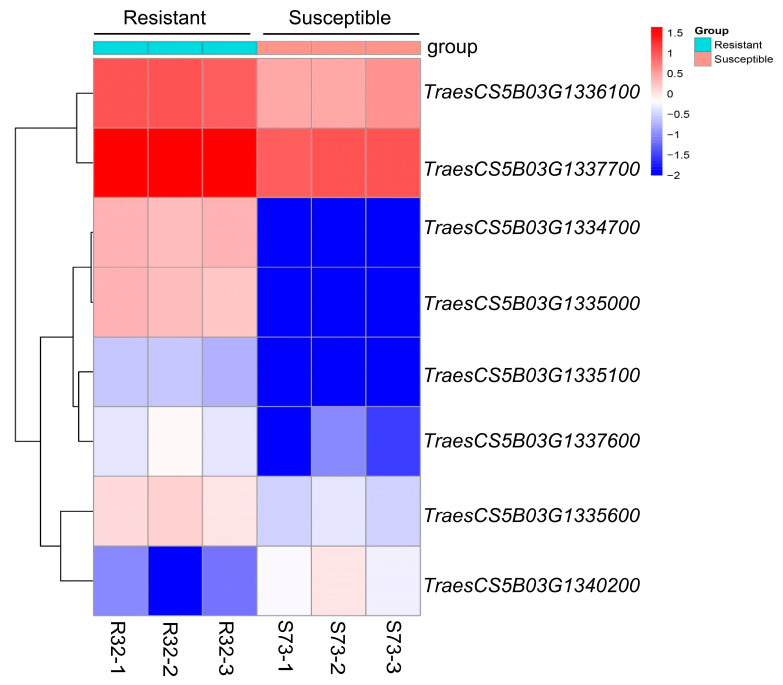
A heat map illustrating the differentially expressed genes (DEGs) within the candidate region of *SrPI94701*. DEGs between homozygous resistant plants (R32-1, R32-2, and R32-3; R) and homozygous susceptible plants (S73-1, S73-2, and S73-3; S) were identified from RNA-seq analysis of *Pgt*-inoculated RNA samples from R32 and S73. The heatmap was generated using the pheatmap *R* package. Leaves from S73 and R32 inoculated with race 34MKGQM were sampled at 6 days post-inoculation (dpi). Each genotype was evaluated using three independent biological replicates.

**Table 1 plants-13-02197-t001:** Primers used in this study. Band sizes correspond to the PI 94701 allele without digestion. CAPS, Cleaved Amplified Polymorphic Sequence. Ann. T., annealing temperature.

Markers	Marker Type	Forward Primer (5′-3′)	Reverse Primer (5′-3′)	Enzyme	Expected Size (bp)	Ann. T. (°C)
*pku68299*	CAPS	GGTTTTAGTGCTGCACCTGGAC	GGCTCTCAGTTCTCTTCTGCACC	HphI	337	63
*pku68425*	CAPS	ACAGACCCCCTTAAGCCTTTTTCTT	AGGGGAGATGTGTGTTGCTTTGTGT	BssHII	441	60
*pku68823*	CAPS	ACTCCTACGGATCAAATTATCACCTT	GCACGGACATCTTGCTAGTAAGAG	ApoI	473	56
*pku69013*	CAPS	CATAATCTTGACGATCCAGGGAC	TATATGCAGGCTATTACTGCTGTGG	HaeIII	609	58
*pku69020*	CAPS	CCAGTTTTTATCGTCCAAATCTAGAG	ATCCATAGGTAGCTGCACATGT	HaeIII	511	54
*pku69118*	CAPS	GTATGAAACCGCGAACACTTTACA	CGGGTTTCCAAATTTTGTTCTTGAG	XmnI	394	57
*pku69119*	CAPS	GGAATTTCACATTTGTTCCCAATC	CGGAGATCGTCAACATCTC	HhaI	394	55
*pku69124*	CAPS	TCTTTGTATTAAGAGTTTGCACAGCT	GCAGATTTCACATACTCAACCATC	SspI	376	57
*pku69187*	CAPS	GCGCTGATGAAGATAATCTCAT	CGGAGGGAGTACTAGATTATCATG	BbvCI	526	57
*pku69211*	CAPS	ATTTGTGTTCATCGATCAAAACAC	TAGTAAGATAAACTCTTGCCTCCTTC	HpyCH4IV	376	52
*pku69227*	CAPS	GGCACCTTTAAAATAATACACGGA	AATGAGTTTGTTGTACCAAGTGCAG	PvuII	354	55
*pku69228*	CAPS	CCTTCCCTACGGATATGTTTTTAGA	AGAAGTTGGAAGGGTAGATCATCACC	Hpy188III	386	55
*pku69231*	CAPS	TGACACTTTCCACTCACTCCTAGG	ATTTGGCACGTTGACCTTAACT	BtgI	340	56
*pku69264*	CAPS	AAATTCTATCAACACTTGAAGAGAA	CCAACCAACTATCATTTAGAAGT	BstUI	503	52
*pku69384*	CAPS	ACTCCTTCACGCTTCTCGACA	AAATTTCCTGGGTGAGCCATT	BanI	496	56
*pku69400*	CAPS	GGTGGTGGAGAACATGCATGC	ATGGCGATGACCGTGCAAGG	MspI	336	60
*pku69560*	CAPS	CGTGGTCCGTTTCTCAGAAGA	CGGGAACAGAAGACACACTATATTT	BfuAI	350	56
*pku69883*	CAPS	GTTCATGTTGTTGAGAAGCTAGAC	CACCTTACAAACAAGTGGTCAAC	BsmAI	600	55

## Data Availability

The raw sequencing data reported in this study are archived at the National Genomics Data Center under BioProject accession number PRJCA027291. Data supporting the findings of this study are within the manuscript or the Appendix A.

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
