# Peer review of "Mapping and Candidate Gene Analysis of an All-Stage Stem Rust Resistance Gene in Durum Wheat Landrace PI 94701"

_plants, 2024, doi:10.3390/plants13162197_

Round 1
Reviewer 1 Report
Comments and Suggestions for Authors
Li et al submitted a manuscript titled "Mapping and candidate gene analysis of an all-stage stem rust resistance gene in durum wheat landrace PI 94701" for publication in MDPI Plants. However, it could be more appropriate in MDPI Agronomy, or Agriculture.
All figure descriptions must include a brief description of methods used to generate that part of the results presented in figures. This applies to all figures here.
Fig. 1, Please move Pgt race names in the figure from overlapping the top of pictures, and label them in black font against a white background, similar to RP/SP.
Fig. 2 Mark the red arrow (pointing up) below the X-axis, for better visibility.
Author Response
Li et al submitted a manuscript titled "Mapping and candidate gene analysis of an all-stage stem rust resistance gene in durum wheat landrace PI 94701" for publication in MDPI Plants. However, it could be more appropriate in MDPI Agronomy, or Agriculture.
Author’s Answer: Thank you for taking the time to evaluate our manuscript. We greatly appreciate your positive feedback. In this study, we mapped and characterized an all-stage stem rust resistance gene in durum wheat landrace PI 94701, and analyzed the candidate genes within the mapping region. Our work aligns with the focus of the Special Issue “Mining and Mapping of Disease-Resistant Genes in Wheat” in Plants. We believe our research will garner significant interest within the wheat breeding and disease resistance communities.
All figure descriptions must include a brief description of methods used to generate that part of the results presented in figures. This applies to all figures here.
Answer: Corrected as suggested. We have added descriptions of the methods in the figure legends.
Fig. 1, Please move Pgt race names in the figure from overlapping the top of pictures, and label them in black font against a white background, similar to RP/SP.
Answer: Corrected as suggested. We have moved the Pgt race names to the bottom of the figure.
Fig. 2 Mark the red arrow (pointing up) below the X-axis, for better visibility.
Answer: Corrected as suggested.
Reviewer 2 Report
Comments and Suggestions for Authors
Comments and Suggestions for Authors
Dear Author,
I have an honor to review the manuscript entitled “Mapping and candidate gene analysis of an all-stage stem rust resistance gene in durum wheat landrace PI 94701” a research article submitted to MDPI Journal, Plants. Authors of this manuscript mapped and characterized an all-stage Sr gene in wheat and mapped the SrPI94701 gene within a 0.17-cM region and identified eight genes within the region. To characterize, they have performed a series of research using this gene. Overall, the experiments are performed well and the results are convincing. Thus, the presented results take up an important topic consistent with the profile of the Journal.
-However, even, manuscript is well organized and well described of the conception, I have some suggestions, which might improve the manuscript to make important to the wider audience.
-Major comments
-Firmed aim of the study that should be underlined precisely and simultaneously and highlight why this gene mapping and analysis is important to study in wheat.
-There are many places where grammar can be improved. I suggest a careful revision by a professional language editing service.
Abstract: -Good organization with results order.
Introduction:
-Introduction is not straightforward relating to results. Many unnecessary descriptions also highlighted. Need substantial improvement
2. Results
- Fig. 1; should have control picture having no pathogen challenged
-L180; which part and which age of plants were inoculated with pathogen?
3. Discussion
This is good discussion for the journal Plants. However, I suggest, some improvement having results comparing with recent article in other plants related to concerned and similar genes functions.
4. Methods
-How the plants were challenged with Pgt race 34MKGQM?
-When the research was performed and what was the cultivation condition.
-sampling procedure should be clearly described
5. Conclusions
Conclusion is too scanty. Make some conclusive remark with results obtained.
pleae check pdf where I made some corrections.

Author Response
I have an honor to review the manuscript entitled “Mapping and candidate gene analysis of an all-stage stem rust resistance gene in durum wheat landrace PI 94701” a research article submitted to MDPI Journal, Plants. Authors of this manuscript mapped and characterized an all-stage Sr gene in wheat and mapped the SrPI94701 gene within a 0.17-cM region and identified eight genes within the region. To characterize, they have performed a series of research using this gene. Overall, the experiments are performed well and the results are convincing. Thus, the presented results take up an important topic consistent with the profile of the Journal.
-However, even, manuscript is well organized and well described of the conception, I have some suggestions, which might improve the manuscript to make important to the wider audience.
Answer: Thank you very much for recognition of our work.
please check pdf where I made some corrections.
- Line3, PI 94701, no need.
Answer: We respectfully disagree with the reviewer here. The resistance gene was identified in PI 94701.
- Line24, in which chromosome
Answer: We added “on chromosome arm 5BL” in the Abstract.
- Line28, the full name of NLR
Answer: We have added full name for abbreviation: nucleotide-binding leucine-rich repeat (NLR) genes
- Lines31, what is uncharacterized wheat genotypes?
Answer: We changed to “in a survey of 104 wheat accessions.”
- Line108, mention how did you check these 73 segregating
Answer: These 73 segregating families were identified through phenotyping. We rephrased the sentences, as follows: “Among the 143 F2:3 families evaluated with Pgt race 34MKGQM, 32 were homozygous resistant, 73 were segregating, and 38 were homozygous susceptible.”. Approximately 25 F3 plants from each F2:3 family were challenged with Pgt race 34MKGQM. If there was a segregation of resistant and susceptible individuals among the 25 seedlings, the family was considered segregating (heterozygous).
- Line 125, should have control picture having no pathogen challenged
Answer: Corrected as suggested.
- Line393, DEGs??? Line390, elaborate dpi
Answer: See line208, where we spelled out “differentially expressed genes (DEGs)” upon its first appearance.
See line239, where we spelled out "days post-inoculation (dpi)" upon its first appearance.
- Line 407, incomplete
Answer: We have improved the conclusion and made the suggested changes, as follows: “In this study, we mapped and characterized SrPI94701, an all-stage stem rust resistance gene in durum wheat landrace PI 94701. The broad efficacy of SrPI94701 in China makes it a desirable target for introgression into Chinese bread wheat cultivars. For recurrent wheat cultivars with different marker haplotypes, the flanking and fully linked markers for SrPI94701 can be used to transfer this gene from durum into modern bread wheat varieties. While SrPI94701 confers only partial resistance to Pgt races, this might be beneficial in breeding programs aiming to combine multiple partial resistance genes and avoid major all-stage resistance genes, a strategy that has been proposed to increase the durability of wheat resistance to stem rust [36].
Using a large mapping population consisting of 2,302 recombinant gametes, SrPI94701 was successfully mapped to the telomeric region of chromosome arm 5BL within a 0.17-cM region. The presence of NLR genes in the collinear regions of wheat reference genomes indicates that SrPI94701 may be a typical NLR gene. However, final identification of the SrPI94701 gene will be required to fully test this hypothesis and determine the relationship between SrPI94701 and Sr49. To ascertain if the two identified NLR candidate genes are required for resistance to Pgt, we are currently conducting functional characterization using truncation mutations for each gene.
In summary, the high‑resolution genetic map of SrPI94701 and the closely linked PCR markers developed in this study will facilitate map-based cloning of this Sr gene and accelerate its incorporation into modern wheat breeding programs.”
-Major comments
-Firmed aim of the study that should be underlined precisely and simultaneously and highlight why this gene mapping and analysis is important to study in wheat.
Answer: Mapping and characterizing the gene SrPI94701 is critical for wheat improvement because stem rust is a devastating disease, causing significant yield losses in wheat worldwide. To reduce losses in wheat, new resistance genes are urgently needed to combat the evolving pathogen population. Identifying and understanding the genetic basis of resistance is therefore crucial for the development of durable resistance strategies.
Marker-assisted selection with closely linked markers can facilitate the efficient introgression of this gene into elite wheat varieties. SrPI94701 can serve as a valuable resource for wheat breeding programs to enhance disease resistance.
By pinpointing the location of SrPI94701 and understanding its genetic architecture, we can facilitate map-based cloning of this Sr gene and accelerate the development of wheat varieties with improved resistance to stem rust, thereby contributing to global food security.
-There are many places where grammar can be improved. I suggest a careful revision by a professional language editing service.
Answer: The manuscript language has been refined through collaboration with a native speaker.
Abstract: -Good organization with results order.
Introduction: -Introduction is not straightforward relating to results. Many unnecessary descriptions also highlighted. Need substantial improvement
Answer: Thank you for your suggestion. We have made all the necessary changes.
- Results
- Fig. 1; should have control picture having no pathogen challenged
Answer: Corrected as suggested. Please refer to the revised Figure 1
-L180; which part and which age of plants were inoculated with pathogen?
Answer: In the Materials and Methods section, lines 348-353, we mentioned that the seedlings were inoculated at the three-leaf stage and whole plants. As follows: “Subsequently, a urediniospore/talc suspension was applied to seedlings at the three-leaf stage using the shaking off method [54]. Following inoculation, the plants were kept in a dark dew chamber overnight and then transferred back to growth chamber conditions maintained at 22–24 °C with a 16-h photoperiod. Pgt infection types (ITs) were recorded 12-14 dpi using the Stakman 0–4 scale indicating levels of immunity to susceptibility [55].”.
- Discussion
This is good discussion for the journal Plants. However, I suggest, some improvement having results comparing with recent article in other plants related to concerned and similar genes functions.
Answer: We appreciate the reviewer's suggestion to include comparative analyses with recent studies of related genes in other plant species. However, our discussion focuses primarily on the novel results within our study system. We will carefully consider whether we can include relevant data from other plant species to strengthen our discussion.
- Methods
-How the plants were challenged with Pgt race 34MKGQM?
Answer: The procedure for challenging seedlings with Pgt race 34MKGQM is described in lines 343-354: “Six Chinese and North American Pgt races, namely BCCBC, 34MTGSM, 34C3RTGQM, 21C3CTTTM, 34MKGQM, and 34C3RKGQM [53], were evaluated for virulence against parental lines PI 94701 and Rusty. The avirulence/virulence profiles of these Pgt races are detailed in Supplementary Table S1. Pgt urediniospores, stored at − 80 °C, were subjected to heat shock at 42 °C for 3 minutes. Subsequently, a urediniospore/talc suspension was applied to seedlings at the three-leaf stage using the shaking off method [54]. Following inoculation, the plants were kept in a dark dew chamber overnight and then transferred back to growth chamber conditions maintained at 22–24 °C with a 16-h photoperiod. Pgt infection types (ITs) were recorded 12-14 dpi using the Stakman 0–4 scale indicating levels of immunity to susceptibility [55]. For plants carrying recombination events within the candidate region, progeny tests were conducted using approximately 25 plants from each F2:3 family.”
-When the research was performed and what was the cultivation condition.
Answer: The time point and cultivation conditions are described in lines 348-353: “Subsequently, a urediniospore/talc suspension was applied to seedlings at the three-leaf stage using the shaking off method [54]. Following inoculation, the plants were kept in a dark dew chamber overnight and then transferred back to growth chamber conditions maintained at 22–24 °C with a 16-h photoperiod. Pgt infection types (ITs) were recorded 12-14 dpi using the Stakman 0–4 scale indicating levels of immunity to susceptibility [55].”
-sampling procedure should be clearly described
Answer: The sampling procedure is described in lines 356-360. To be more specific, we added several sentences in the text: “Based on the phenotypic data, 15 homozygous resistant and 15 homozygous susceptible F3 families were selected to constitute the resistant bulk (R-bulk) and susceptible bulk (S-bulk) for RNA extraction. Leaves of equal length were taken from the second leaf of all seedlings in each family. After harvest, each sample was immediately flash-frozen in liquid nitrogen and stored at -80°C for RNA extraction.”
- Conclusions
Conclusion is too scanty. Make some conclusive remark with results obtained.
Answer: We have improved the conclusion and made the suggested changes.